# Multi-Layered Projected Entangled Pair States for Image Classification

**Lei Li and Hong Lai \***

School of Computer and Information Science, Southwest University, Chongqing 400715, China
\* Correspondence: hlai@swu.edu.cn

**Abstract:** Tensor networks have been recognized as a powerful numerical tool; they are applied in various fields, including physics, computer science, and more. The idea of a tensor network originates from quantum physics as an efficient representation of quantum many-body states and their operations. Matrix product states (MPS) form one of the simplest tensor networks and have been applied to machine learning for image classification. However, MPS has certain limitations when processing two-dimensional images, meaning that it is preferable for an projected entangled pair states (PEPS) tensor network with a similar structure to the image to be introduced into machine learning. PEPS tensor networks are significantly superior to other tensor networks on the image classification task. Based on a PEPS tensor network, this paper constructs a multi-layered PEPS (MLPEPS) tensor network model for image classification. PEPS is used to extract features layer by layer from the image mapped to the Hilbert space, which fully utilizes the correlation between pixels while retaining the global structural information of the image. When performing classification tasks on the Fashion-MNIST dataset, MLPEPS achieves a classification accuracy of 90.44%, exceeding tensor network models such as the original PEPS. On the COVID-19 radiography dataset, MLPEPS has a test set accuracy of 91.63%, which is very close to the results of GoogLeNet. Under the same experimental conditions, the learning ability of MLPEPS is already close to that of existing neural networks while having fewer parameters. MLPEPS can be used to build different network models by modifying the structure, and as such it has great potential in machine learning.

**Keywords:** tensor networks; image classification; multi-layered projected entangled pair states

## 1. Introduction

Due to the development of the internet, different forms of massive datum have been generated, resulting in the need for big data analysis methods to be formed. Systems have been developed to make processing these large amounts of data possible [1]. Among big data sources, multimedia data make up the majority [2]. In order to obtain the useful information from image data, computer vision has become an increasingly important method [3]. Image classification is a fundamental method in computer vision for classifying images into one of several predetermined classes, forming the foundation for computer vision tasks such as image segmentation [4]. From manually labeling features for classification to deep learning, many image classification algorithms have been developed [5–9]. Tensor networks, which originate from quantum physics, are gradually being applied to image classification tasks [10–13].

Tensor networks are representations of quantum many-body states based on their local entanglement structures [14]. A tensor network expresses higher-order tensors as multiple low-order tensors in the form of contractions [15], thereby avoiding the "curse of dimensionality" problem [16]. These lower-order tensors can be combined in different ways to obtain different tensor networks. After the typical datasets are embedded into a quantum state, they plays a role analogous to that of the area law in quantum physics [17,18]. This allows tensor networks to be gradually introduced into image classification [10,19–21].

Tensor networks based on MPS have not achieved very good results in image classification tasks due to the loss of image structure information [10,20]. In order to better obtain the global information of the image, the PEPS tensor network [20], which uses the same geometric structure as the natural image, has been introduced for image classification; PEPS can directly obtain the structural information of the whole image, although it sometimes ignores the correlation between local pixels. In deep learning, great achievements have been made through the hierarchical method. For example, after the convolutional neural network [22–24] extracts abstract features through more layers of convolution, the fully connected layer can achieve better classification results. The idea of hierarchical methods has been introduced for tensor networks as well [11,13,25].

In this work, we use the hierarchical method to build a hierarchical PEPS tensor network that can obtain the global information of the image without losing the local correlation between pixels; we call this the MLPEPS tensor network. MLPEPS is composed of multiple PEPS layers, with each PEPS layer potentially being composed of multiple PEPS blocks. All PEPS blocks in each PEPS layer have the same structure, and the PEPS layer structure can be changed by modifying the size of the PEPS block, making the model more variable. The lower-layer PEPS is used to obtain the local abstract information of the image, while the upper-layer PEPS classifies the images according to the output of the lower-layer PEPS. We use the boundary MPS method [26] to approximately contract all PEPS blocks in MLPEPS, and all parameters are optimized by backpropagation algorithm [27]. We verified the learning ability of MLPEPS using the Fashion-MNIST dataset, on which its test set accuracy reached 90.44%. In comparison with the existing tensor network model, the test set accuracy of MLPEPS surpassed all other tensor networks and the AlexNet convolutional neural network. On the COVID-19 Radiography dataset, the test set accuracy of MLPEPS was much higher than that of single-layer PEPS. Moreover, MLPEPS can obtain results close to those of mature neural networks while requiring fewer parameters. Our experiments have shown that while MLPEPS has strong learning ability, its generalization performance needs to be improved. A wide variety of MLPEPS models can be formed by changing the basic PEPS block structure, which greatly compensates for the relatively poor scalability of previous tensor network models.

The rest of this paper is organized as follows. Section 2 introduces the relevant research on neural networks and tensor networks in image classification. Section 3 presents methods for performing image classification using tensor networks. Section 4 details the method for constructing MLPEPS. Section 5 introduces the results of MLPEPS on image classification and compares them with those of other methods. Finally, Section 6 concludes the paper and discusses more ways to use MLPEPS.

## 2. Literature Review

The construction of the ImageNet [28] project has promoted the development of image recognition technology and has become the standard for image classification algorithm evaluation. Currently, convolutional neural networks are the most effective method for image classification tasks, and there are many useful algorithms. A major development in deep learning, AlexNet [29] promoted the key step of neural network from shallow layer to deep layer, and proved that abstracting features layer by layer through a hierarchical method can obtain better classification performance than manually extracting features. VGGNet [9] explored the impact of depth on the accuracy of convolutional neural networks for image classification tasks, proving that deeper networks can achieve better classification results on ImageNet. By combining convolution kernels of different sizes, GoogLeNet [5,6,30] improved the accuracy of classification while reducing computational complexity. GoogLeNet achieves good performance thanks to its hierarchical network structure. ResNet [7] and DenseNet [31] have both been able to achieve better classification accuracy by constructing residual blocks and dense blocks. This allows them to precisely use the hierarchical method to build a deeper convolutional network model. Based on these developments, deep convolutional neural networks uses the hierarchical methods to achieve great achievements. This

demonstrates the validity, and even the necessity, of using a hierarchical method. Although deep convolutional neural networks have achieved good results in image classification tasks, many problems arise as well; for example, better performance requires very deep networks, which lead to excessive parameter volume and computational complexity. At the same time, with increasing network depth various problems begin to appear in the optimization algorithm of the model, such as gradient disappearance and gradient explosion [32]. Therefore, methods that are wholly different from convolutional neural networks have been gradually introduced into image classification; for instance, transformer-based methods have been widely used in image classification recently, achieving a great success [33]. The tensor network method originating from quantum physics has been widely used in image classification due to its own advantages [10].

An MPS composed of third-order tensors is the simplest tensor network used as a machine learning model for image classification [10]. By performing classification on the MNIST dataset, MPS has demonstrated the feasibility of tensor networks for image classification. The main method is to use MPS for classification after mapping the data to the Hilbert space. Many MPS-based tensor networks have been applied to image classification. Generative tensor network classification (GTNC) [21] is a generative MPS tensor network model that achieves 98.2% test accuracy on the MNIST dataset by learning the probability distribution of the dataset. The Locally-orderless Tensor Network (LoTeNet) [25] is a hierarchical MPS tensor network that uses MPS to extract useful features from locally orderless small regions in images, enabling larger images to be handled without losing information on the global structure. Multi-layered tensor network (MLTN) [11] is a powerful supervised learning model that consists of hierarchical MPS blocks and implements a linear classifier in high-dimensional space. The reported literature indicates that the success of these tensor networks is mostly due to the introduction of hierarchical methods, proving that the usefulness and power of such methods extends to tensor networks. However, when using MPS-based approaches for image classification, the image needs to first be converted into a one-dimensional vector, which can lead to loss of global structural information. To avoid this issue, tensor networks that can obtain image structure information have been proposed and applied to image classification.

The 2D Multi-scale Entanglement Renormalization Ansatz (MERA) [12] model is a tensor network that is essentially a cascade of isometries and disentanglers. This model can be directly applied to two-dimensional images; it is able to obtain coarse-grained data layer-by-layer through the hierarchical structure and obtain more abstract information while retaining image structure information. It uses the entanglement entropy theory to achieve results similar to a CNN in tiny object classification tasks. A Tree Tensor Network (TTN) [13] uses 2D unitary tensors to form a hierarchical tensor network similar to MERA. Using a layer-by-layer feature abstraction method similar to a deep convolutional neural network, TTN achieves superior performance on two-dimensional image datasets. A Deep Convolution Tensor Network (DCTN) [34] uses tensors instead of convolution kernels, which greatly reduces the amount of parameters required by the model. Its performance is affected by the network depth. Even for these two-dimensional tensor networks, performance gains result from using the hierarchical method.

The PEPS [20] tensor network uses the advantages of its own structure to obtain state-of-the-art performance on the MNIST and Fashion-MNIST datasets, which proves the advantages of two-dimensional tensor networks. Although this tensor network has achieved great success, there remains a large gap in its image classification performance compared to mature neural networks. This paper combines the advantages of PEPS and the hierarchical method in an effort to further reduce the gap between tensor networks and neural networks; our experimental results prove that MLPEPS can achieve the performance on par with complex neural networks.

### 3. Tensor Network for Image Classification

Supervised learning is a very common method in machine learning, and has been widely used for a variety of tasks [10]. Image classification is a simple supervised learning task. In image classification tasks, each data sample $x$ contains a ground-truth label $y \in \{1, 2, \cdots, T\}$ to represent different classes of data. For a grayscale image sample $x \in \mathbb{R}^{L_0 \times L_0}$, the main purpose of image classification is to learn a classifier function $f$ that makes $y^{pred} = f(x)$ closer to $y$.

The main method of tensor network image classification is to use the tensor network as the classifier function. The prediction results $y^{pred}$ are obtained by contracting the entire tensor network and the image feature vector. Various tensor network models have been used as classifiers in image classification, such as MPS [10], TTN [13], and PEPS [20]. When the tensor network is used as a classifier, an output index is included as the prediction result, as shown in Figure 1.

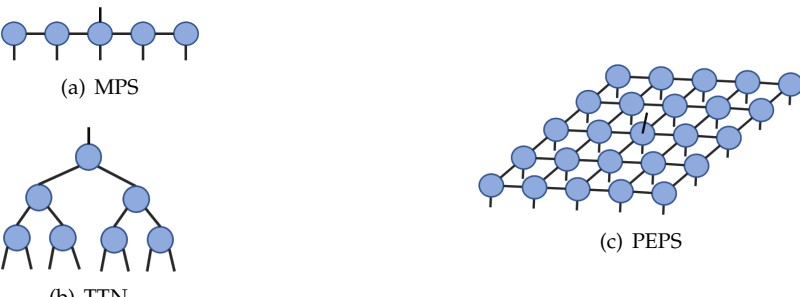

(a) MPS

(b) TTN

(c) PEPS

**Figure 1.** (**a**) MPS, (**b**) TTN, and (**c**) PEPS. The upward index in each tensor network represents the output index, while the downward index is the physical index.

For an image $x \in \mathbb{R}^{L_0 \times L_0}$ consisting of $N = L_0 \times L_0$ pixels, each pixel is mapped to a feature vector of the Hilbert space through a feature map $\phi(x_i) \in \mathbb{R}^d (i = 1, 2, \cdots, N)$ prior to classification using a tensor network, where $d$ is the dimension of feature vector [18]. The feature vector of all pixels contracts with the physical indices of the tensor network, as shown in Figure 2b. The prediction result is provided by the output index. The feature map of the full image can be defined as

$$\mathbf{\Phi}(x) = \phi(x_1) \otimes \phi(x_2) \otimes \cdots \otimes \phi(x_N). \tag{1}$$

A simple feature map function with $d = 2$ is

$$\phi(x_i) = \begin{pmatrix} \cos\left(\frac{\pi x_i}{2}\right) \\ \sin\left(\frac{\pi x_i}{2}\right) \end{pmatrix} \in \mathbb{R}^2. \tag{2}$$

The feature map transforms an $L_0 \times L_0$ image to an $L_0 \times L_0 \times d$ tensor, as shown in Figure 2a.

We denote the tensor network classification model by $W$; then, the predicted label of the image can be defined as

$$f(x) = W \cdot \mathbf{\Phi}(x). \tag{3}$$

After obtaining the predicted label for the image, the cost function is used to measure the distance between the predicted label $f(x)$ and the ground-truth label $y$, as follows:

$$\mathcal{L} = \frac{1}{M} \sum_{i=1}^{M} l(f(x_i), y_i) \tag{4}$$

where $y_i$ is the ground truth for image $x_i$, $M$ is the total number of images in the dataset, and $l$ is the loss function.

The weight tensor $W$ is randomly initialized; the cost can be optimized using the stochastic gradient descent algorithm to obtain the parameter of $W$, which minimizes the cost function.

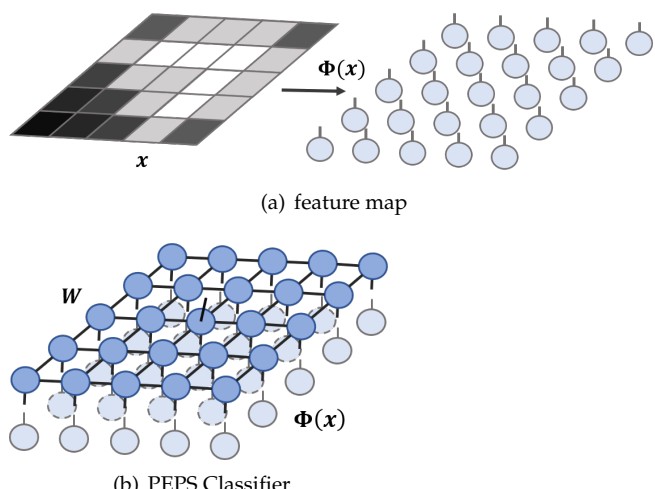

(a) feature map

(b) PEPS Classifier

**Figure 2.** (**a**) Each pixel in the image receives a feature vector through feature map; (**b**) the physical indices of PEPS contract with each feature vector. All of the indices that connect adjacent tensors are virtual indices.

## 4. MLPEPS Classifier

### 4.1. PEPS Classifier

The PEPS classifier is a two-dimensional tensor network supervised learning model [20] with a similar structure to that in the image above. When using PEPS for image classification tasks, $W$ denotes the PEPS tensor network, as shown in Figure 2b; this can be represented using a composition of tensors $A^{[i]}$, as follows:

$$W^{T,s_1,s_2,\cdots,s_N} = \sum_{\sigma_1\sigma_2\cdots\sigma_K} A^{s_1}_{\sigma_1,\sigma_2} A^{s_2}_{\sigma_3,\sigma_4,\sigma_5} \cdots A^{s_i,T}_{\sigma_k,\sigma_{k+1},\sigma_{k+2},\sigma_{k+3}} \cdots A^{s_N}_{\sigma_{K-1},\sigma_K} \tag{5}$$

where $T$ represents the output index, $s_i$ denotes the physical indices, and $\sigma_k$ denotes the virtual indices.

In Figure 2b, the upward index in $W$ represented by PEPS is the output index $T$. All of the downward indices that contract with $\Phi(x)$ are the physical indices $s$, and all of the indices that connect adjacent tensors in $W$ are the virtual indices $\sigma$. All virtual indices contract with the corresponding virtual indices of their adjacent tensors.

Then, the predicted label can be written as

$$f^{[T]}(x) = W^{T,s_1,\cdots,s_N} \cdot \phi_{s_1}(x_1) \otimes \cdots \otimes \phi_{s_N}(x_N). \tag{6}$$

The physical indices $s_i$ in PEPS contract with their corresponding feature vectors. The result of this contraction is a T-dimensional vector, where the $i$-th element in the vector represents the probability that a given image $x$ belongs to class $i$. The index corresponding to the element with the largest value in the output vector is used as the final prediction class result.

### 4.2. MLPEPS Classifier

In this work, we introduce the hierarchical method [11,25] into the PEPS tensor network to obtain an MLPEPS tensor network classifier. While maintaining the natural structure of the image, it obtains more abstract features of the image through MLPEPS, which has better performance than single-layer PEPS.

As shown in Figure 3, MLPEPS consists of multiple PEPS layers. Each PEPS layer consists of multiple identical PEPS blocks that form a square lattice. The output vectors of all PEPS blocks in the lower layers are used as the input vectors of the PEPS block in the upper layer. The input data pass through each PEPS layer in turn, and the output index of the single highest-layer PEPS is used as the prediction result. As in single-layer PEPS, the result is a vector and the index of the largest element is the predicted class.

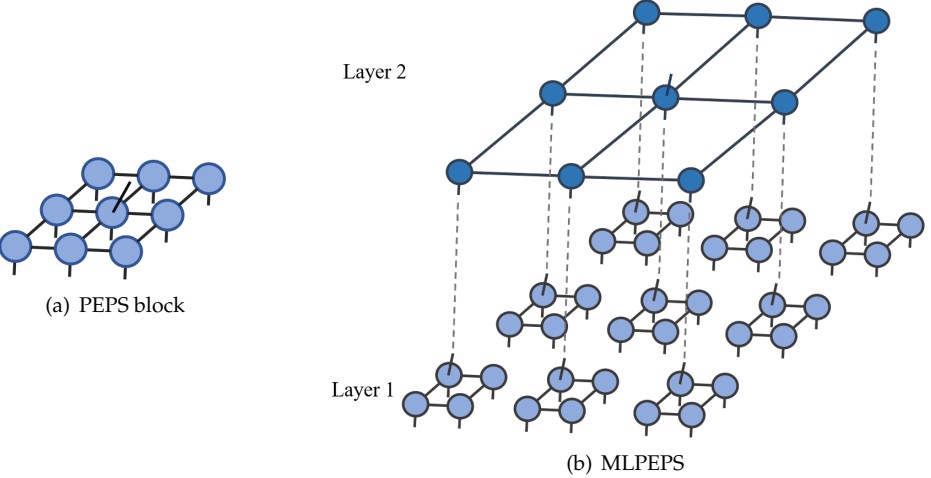

**Figure 3.** (**a**) PEPS block of size $m = 3$ and (**b**) two-layer PEPS classifier. Each layer consists of several PEPS blocks. Each PEPS block contains an output index. The output vectors of the lower-layer PEPS are used as the input vector of the upper-layer PEPS. In this figure, $L_1 = 6$, $L_2 = 3$; $m_1 = 2$, $m_2 = 3$; $N_1 = 9$; $N_2 = 1$.

In MLPEPS, all PEPS blocks in the same PEPS layer have the same structure, meaning that we can express the overall structure as a set of variables. For the $i$-th layer of $H$-layer MLPEPS, the size of each PEPS block is termed $m_i$ and the number of input vectors is $L_i \times L_i$; thus, the number of PEPS blocks $N_i$ in this layer is $L_i/m_i \times L_i/m_i$. Figure 3b shows a two-layer PEPS along with the values of various parameters. In addition, we denote the physical index dimension of each PEPS block as $p_i$ and the output index dimension of each PEPS block as $o_i$. Because the output vector of each lower-layer PEPS block is used as the input vector of the adjacent upper-layer PEPS block, the physical index dimension in the upper layer PEPS block should be equal to the output index dimension of the adjacent lower-layer PEPS block; that is, $p_{i+1} = o_i$. Furthermore, the number of input vectors of the upper-layer PEPS is the same as the number of adjacent lower-layer PEPS blocks; that is, $L_{i+1} \times L_{i+1} = L_i/m_i \times L_i/m_i$.

For the first PEPS layer, the number of input vectors $L_1 \times L_1$ and the dimension $p_1$ of physical indices in the PEPS block depend on the size $L_0 \times L_0$ of the input image and the dimension $d$ of the feature map. In order to obtain more region information, we first apply a squeeze operation [25] to the input image. The squeeze operation is intended to sequentially divide the image into pixel blocks of size $k \times k$. After performing feature mapping on each pixel in a pixel block, the tensor product of the feature vectors of all pixels is used to obtain the feature vector representing the pixel block. If the dimension of the feature map is $d$, for a pixel block of size $k \times k$ the dimension of the feature vector of the pixel block is $d^{k \times k}$. Therefore, for an image of size $L_0 \times L_0$, pixel blocks $L_0/k \times L_0/k$ are obtained after the squeeze operation. The feature vector of each pixel block contracts with the physical index of the PEPS block, meaning that the number of input vectors of the first PEPS layer is $L_1 \times L_1 = L_0/k \times L_0/k$ and the dimension of the physical index of each PEPS block is $p_1 = d^{k \times k}$. The squeeze operation is not required. When the squeeze operation is not performed, it is equivalent to $k = 1$. In this case, $L_1 \times L_1 = L_0 \times L_0$, $p_1 = d$.

It is worth noting that for other PEPS layers there is no squeeze operation or feature map; thus, as mentioned above, $p_{i+1} = o_i$.

### 4.3. Model Optimisation

In MLPEPS, the $W$ parameter contains all PEPS blocks; here, we optimize the parameters by minimizing the cross-entropy loss function composed of the predicted labels and the ground-truth labels:

$$\mathcal{L} = -\frac{1}{M} \sum_{i=1}^{M} \log \left[ \text{softmax} \left( f^{[y_i]}(\boldsymbol{x}_i) \right) \right] \tag{7}$$

where

$$\text{softmax} \left[ f^{[y_i]}(\boldsymbol{x}_i) \right] = \frac{e^{f^{[y_i]}(\boldsymbol{x}_i)}}{\sum_{t=1}^{T} e^{f^{[t]}(\boldsymbol{x}_i)}} \tag{8}$$

with $f^{[t]}$ denoting the $t$-th element in the output vector.

The entire MLPEPS contracts each PEPS layer sequentially from bottom to top, finally obtaining a vector $f(\boldsymbol{x})$ representing the result. However, when the size of the PEPS block is too large exact contraction leads to exponential increase of the internal virtual index dimension of the PEPS blocks. Instead, we use the boundary MPS method to approximate exact results in order to avoid this problem [26]. This method regards the top and bottom tensor rows as an MPS and the tensors of the remaining rows as matrix product operator (MPO), meaning that the contraction of the entire PEPS can be regarded as the operators continuously being applied to the MPS. The key to this process is to truncate the dimension of the virtual index to $\chi$ after each MPO is applied to the MPS. By controlling the dimension of $\chi$, the exponential increase of the virtual index can be avoided within the allowable error [35]. In this way, the exponential computational complexity can be reduced to the polynomial level.

Singular Value Decomposition (SVD) is required to truncate the virtual index dimension. When backpropagating SVD through Pytorch, we encountered the problem of numerical instability in the PEPS classifier [20]. To solve this problem, we used a custom SVD backpropagation process [36].

### 5. Experiments

Because image classification is the main application of tensor networks in computer vision, deep learning has realized great achievements in image classification. For better comparison, we primarily validated MLPEPS on image classification tasks. The hardware environment we used is a Xeon(R) Platinum 8255C processor and NVIDIA GeForce RTX 2080Ti 11G, while the memory size is 45G. The experiment used Ubuntu 18.04 and Python 3.8.3 as the basic environment; the development environment is CUDA 11.2, PyTorch 1.11.0, and TorchVision 0.12.0.

To evaluate the capability of MLPEPS for image classification tasks, we conducted experiments on two datasets. The Fashion-MNIST dataset [37] is used to compare single-layer PEPS and MLPEPS in order to illustrate the learning ability of MLPEPS. This can be compared with the performance of existing TN models. In addition, we verify the learning ability of MLPEPS on the COVID-19 Radiography dataset [38,39] and used well-known neural networks in a comparison to verify its generalization ability. Figure 4 shows the flowchart of the experiment.

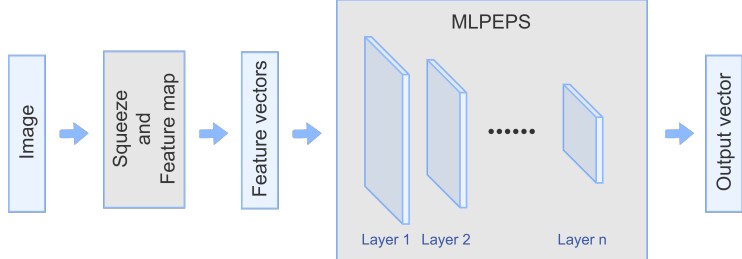

**Figure 4.** Flowchart of MLPEPS for image classification. The feature vectors of the input image are first obtained through a squeeze operation and feature mapping. Afterwards, the feature vectors are contracted by each layer of the MLPEPS in turn, with the output vector of the last layer used as the prediction result.

*5.1. Fashion-MNIST Dataset*

The Fashion-MNIST dataset consists of grayscale images of size $28 \times 28$, and contains ten classes of clothing. There are 60,000 training images and 10,000 test images in the dataset. Example images are shown in Figure 5a.

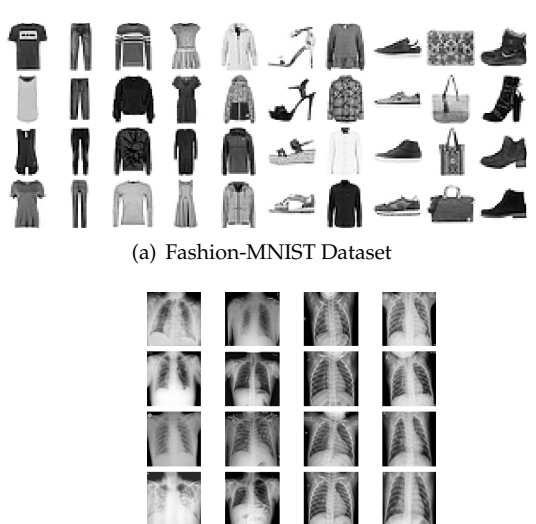

(a) Fashion-MNIST Dataset

(b) Covid-19 Radiography Dataset

**Figure 5.** Example images from the Fashion-MNIST and COVID-19 Radiography datasets. Each column of images belongs to the same class.

Different MLPEPS structures can be formed by modifying the size of the PEPS block and the number of PEPS layers. Because the images in Fashion-MNIST are relatively small, we only build a two-layer PEPS for the experiments. By changing the size of each layer of PEPS blocks, we find that the structure with the best results is the first-layer PEPS block with size $m_1 = 2$ and the second-layer PEPS block with size $m_2 = 7$. The detailed structure is provided below.

The size of all images in Fashion-MNIST is $L_0 \times L_0 = 28 \times 28$; after the $k = 2$ squeeze operation, $14 \times 14$ pixel blocks are obtained. Each pixel block can obtain a $d^{k \times k} = 2^{2 \times 2} = 16$ dimensional feature vector through feature mapping based on Equation (2) and the tensor product, that is, $L_1 \times L_1 = 14 \times 14$, $p_1 = 16$. Each PEPS block of $m_1 = 2$ corresponds to the adjacent $2 \times 2$ feature vectors as input; thus, there are $N_1 = L_1/m_1 \times L_1/m_1 = 14/2 \times 14/2 = 7 \times 7$ PEPS blocks in the first layer.

In the second layer, the number of input vectors is $L_2 \times L_2 = L_1/m_1 \times L_1/m_1 = 14/2 \times 14/2 = 7 \times 7$; because there is only one PEPS block, its size is $m_2 = 7$. The physical index dimension of this PEPS block is the same as the input dimension of the first layer PEPS block, that is, $p_2 = o_1$.

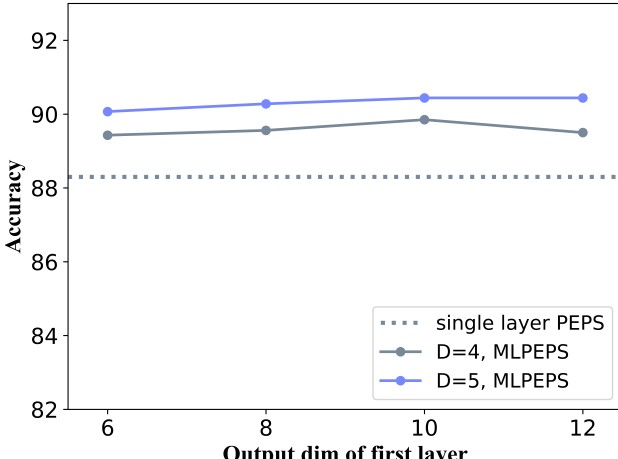

**Figure 6.** Classification accuracy of two-layer PEPS on Fashion-MNIST test dataset when choosing different sizes of $D$ and $o_1$; the test set accuracy exceeds that of the single-layer PEPS under all conditions.

In our experiments, we find that modifying the output index size of the PEPS block $o_1$ in the first layer affects the prediction results. The output index size of the PEPS blocks corresponds to how much lower-layer information is retained; thus, we set $o_1$ as a hyperparameter. The output of the second-layer PEPS is used as the prediction result, and $o_2 = 10$ is set for the Fashion-MNIST experiment.

As the dimension of the virtual index $\sigma$ in each PEPS block has a great influence on the learning ability of the model, we use virtual index dimension $D$ as a hyperparameter. The learning rate is set to 0.0001 in all experiments.

Figure 6 shows a comparison of the test set accuracy with $D$ and $o_1$ selected with different sizes. When $D = 3$, the test set accuracy of two-layer PEPS exceeds that of single-layer PEPS (not shown in the figure). As $D$ increases, the two-layer PEPS sees greater improvement compared to the single-layer PEPS. MLPEPS obtains more abstract features by adding PEPS layers, meaning that better classification results are achieved. In this experiment, we find that the accuracy of the two-layer PEPS on the training set reaches 100% when $D = 5$ and $o_1 \geq 10$, while the single-layer PEPS fails to achieve it under any conditions [20]. This result proves that MLPEPS achieves better learning ability by introducing the hierarchical method.

Table 1 lists the comparison of the highest test accuracy between MLPEPS and other models on the Fashion-MNIST dataset. MPS-based tensor network models, such as MPS and Multi-scale TNs, need to expand the image into one dimension, which may lead to the loss of structural information, so there is a large gap in accuracy compared to MLPEPS. It can also be seen from the results that the accuracy of MLPEPS exceeds that of the classic machine learning model XGBoost and the deep convolutional neural network model AlexNet, which shows the powerful learning ability of MLPEPS. It cannot be ignored that the performance of MLPEPS still has some gaps compared to a more mature neural network such as GoogLeNet. But as a new approach to machine learning, MLPEPS has shown great potential. Because the network structure of MLPEPS has many possibilities, with the attempts of different structural models, a model with stronger learning ability may be obtained.

**Table 1.** Accuracy comparison on the Fashion-MNIST test dataset. MLPEPS outperforms all compared tensor networks as well as a number of other well-known machine learning models.

| Model | Test Accuracy |
| --- | --- |
| **MLPEPS** | **90.44**% |
| PEPS [20] | 88.30% |
| MPS [27] | 88.00% |
| Multi-scale TNs [19] | 88.97% |
| DCTN [34] | 89.38% |
| XGBoost [19] | 89.80% |
| AlexNet [40] | 89.90% |
| **GoogLeNet** [40] | **93.70**% |

*5.2. COVID-19 Radiography Dataset*

The COVID-19 radiography dataset contains four chest X-ray images for 3616 COVID-19 positive cases along with 10,192 normal, 6012 lung opacity (Non-COVID-19 lung infection), and 1345 viral pneumonia images. The size of each image is $299 \times 299$. In order to simplify the experiment, the image size is resized to $32 \times 32$ for our experiment. For comparison, we selected the first 1345 images from each class to form a balanced four-class dataset. In the experiment, all the images are divided into a training set and test set according to a ratio of 8:2.

First, we use a two-layer PEPS for the classification task and set $k = 2$, $m_1 = 2$, $m_2 = 8$. As in the Fashion-MNIST experiment, we find the best MLPEPS hyperparameters by setting $o_1$ and $D$ of different sizes. As a comparison, we conduct the same experiment on single-layer PEPS, GoogLeNet [5], AlexNet [29], and VGG-16 [9]. Due to the limitations of Pytorch, in the AlexNet experiment the image size is set to $63 \times 63$. The learning rate is set to 0.0001 in all experiments, and the cross-entropy loss function is used.

Figure 7 shows the best test set accuracy within 300 epochs for all comparative experiments. The solid line in the figure shows the variation of the test classification accuracy with the change of $D$ and $o_1$ in the single-layer PEPS and two-layer PEPS experiments. It can be seen from the results that the classification accuracy of the two-layer PEPS on the test set far exceeds that of the single-layer PEPS in all cases; indeed, the accuracy of two-layer PEPS with $D = 3$ exceeds that of single-layer PEPS with $D = 5$. In PEPS, the dimensionality of the virtual index $D$ affects the learning ability of the model. The fact that MLPEPS with $D = 3$ far exceeds the performance of single-layer PEPS with $D = 5$ proves the stronger learning ability of two-layer PEPS. It can be seen from the figure that the changes in $o_1$ and $D$ have no great impact on the classification accuracy of MLPEPS, showing that MLPEPS is more robust than that of single-layer PEPS. On complex datasets such as the COVID-19 Radiography dataset, the gap between single-layer PEPS and MLPEPS is further amplified. This proves that MLPEPS can achieve better learning ability than single-layer PEPS by acquiring the abstract image features through a layer-by-layer process.

Table 2 shows the best results on the training and test sets for all models. In these experiments, we find that when $D \geq 3$ the training set accuracy of the two-layer PEPS can reach 100% in all conditions, while when $D \geq 4$ the accuracy of the single-layer PEPS on the training set can reach 100%. However, all neural networks fail to achieve 100% accuracy on the training set, and the training set accuracy of VGG-16 is only 99.07%. This highlights the better fitting ability of tensor networks for data, and proves the potential of tensor networks for machine learning.

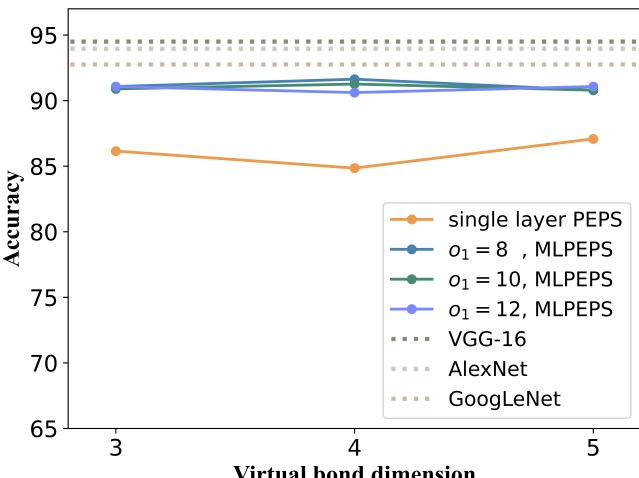

**Figure 7.** Comparison of classification test set accuracy of different models on the COVID-19 Radiography dataset. The horizontal axis indicates the virtual index dimension $D$ of different sizes in the single-layer PEPS and two-layer PEPS experiment. The figure plots the change in classification accuracy of two-layer PEPS for different $o_1$. The classification accuracy of the two-layer PEPS far exceeds that of the single-layer PEPS, and there is only a small performance gap to mature neural networks.

**Table 2.** Comparison of the best training set and test set accuracy of single-layer PEPS, two-layer PEPS, and neural networks. Two-layer PEPS achieves the highest test set accuracy when $o_1 = 8, D = 4$, and its accuracy is very close to that of GoogLeNet. The training set accuracy of two-layer PEPS outperforms the other models, reaching 100% in all cases.

| Model | Train Accuracy | Test Accuracy |
|---|---|---|
| single layer PEPS ($D = 5$) | 100% | 87.08% |
| 2-layer PEPS ($o_1 = 8, D = 4$) | 100% | **91.63**% |
| GoogLeNet | 99.95% | 92.75% |
| AlexNet | 99.72% | 93.95% |
| VGG-16 | 99.07% | **94.50**% |

It can be seen from Table 2 that when $o_1 = 8$, $D = 4$, the test set accuracy of the two-layer PEPS is already very close to GoogLeNet. Figure 8 shows the confusion matrix of the test set classification results at this point. It cannot be ignored that although MLPEPS has better performance on the training set, its test set accuracy has gaps compared to AlexNet and VGG-16 on the COVID-19 radiography dataset. This shows that although MLPEPS has very strong learning ability, its generalization ability remains relatively weak. The accuracy of MLPEPS on the training set reached 100% and surpassed that of the neural networks; however, its accuracy on the test set is far behind that of the neural networks. This is most likely due to overfitting. Compared with neural networks, tensor networks represent a new machine learning model, and there is currently no mature method for avoidance of overfitting. In view of the extant methods for avoiding overfitting in neural networks, this is a problem worth studying in the context of tensor networks, and can help further exploit the huge potential of such networks.

Although neural networks have an advantage in accuracy on the test set compared to MLPEPS, the parameters required by the two-layer PEPS are greatly reduced compared to the neural networks. Table 3 shows a comparison of the number of parameters required by the different models. Two-layer PEPS achieves impressive performance with only 1/90 the parameters of VGG-16. As mentioned above, tensor networks can approximate higher-order tensors through lower-order tensors, meaning that tensor networks can use fewer parameters to represent more information. According to this characteristic, tensor networks have been widely used in low-rank approximation and data compression [41,42].

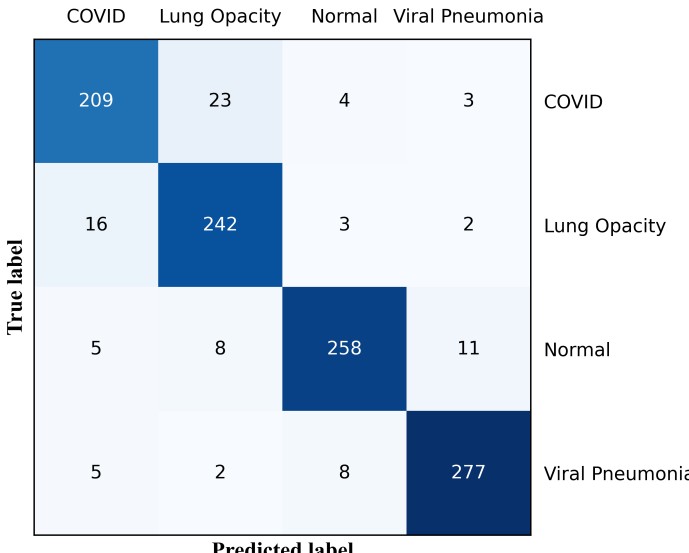

**Figure 8.** Confusion matrix of the classification results of the two-layer PEPS on the test set of the COVID-19 Radiography dataset with $o_1 = 8, D = 4$.

To further verify the learning ability of MLPEPS, we construct a deeper PEPS tensor network for the COVID-19 radiography dataset four-class classification task. For an image size of $64 \times 64$, we construct three-layer, four-layer, and five-layer PEPS networks. The experimental results and hyperparameters of each network are shown in Table 4. It is found that the accuracy of MLPEPS on the test set gradually decreases with increasing depth. Similar to deep learning, an increase in the number of network layers may cause problems with overfitting, gradient disappearance, and gradient explosion, resulting in decreased model performance. Therefore, the impact of a deeper PEPS tensor network model on performance may require further exploration. In addition, the degradation of model performance may be due in part to an inappropriate network structure. Because changing the PEPS block size can lead to completely different model structures, the model we construct in our experiments may not be the best model for the current dataset. These results are provided simply to illustrate the learning ability of MLPEPS. In future attempts, it is possible that more suitable structures for this task may be discovered.

**Table 3.** Comparison of the number of parameters of different models. Two-layer PEPS has a relatively smaller number of parameters; when $D = 3$, it has only 1/90 the amount of parameter of VGG-16. Compared with neural networks, MLPEPS has a great advantage in terms of the number of parameters.

| Model | Parameter | Ratio |
|---|---|---|
| single layer PEPS(D=4) | 1,064,964 | 0.76 |
| 2-layer PEPS($o_1 = 12, D = 3$) | 1,394,102 | 1.00 |
| 2-layer PEPS($o_1 = 12, D = 4$) | 4,404,102 | 3.16 |
| 2-layer PEPS($o_1 = 12, D = 5$) | 10,750,902 | 7.71 |
| GoogLeNet | 5,604,004 | 4.02 |
| AlexNet | 57,020,228 | 40.90 |
| VGG-16 | 134,276,932 | 90.32 |

**Table 4.** Deeper PEPS models are used to compare the test accuracy on the four-class COVID-19 Radiography dataset; the *i*-th element in *m* and *o* represents the block size and output index size, respectively, of the *i*-th layer PEPS.

| Model | *m* | *o* | D | Test Accuracy |
|---|---|---|---|---|
| 3-layer PEPS | $[4, 2, 4]$ | $[6, 6, 4]$ | 4 | 90.33% |
| 4-layer PEPS | $[4, 2, 2, 4]$ | $[8, 8, 8, 4]$ | 4 | 89.12% |
| 5-layer PEPS | $[4, 2, 2, 2, 4]$ | $[8, 8, 8, 8, 4]$ | 4 | 88.84% |

## 6. Conclusions

By extending the single-layer PEPS model to multiple layers, in this paper we have constructed the MLPEPS model for image classification. We use stochastic gradient descent to update the parameters of the model. By abstracting the image information layer-by-layer through multiple PEPS layers, MLPEPS achieves better classification results on image classification tasks compared to single-layer PEPS. In order to verify the learning ability of MLPEPS, experiments are first performed on the Fashion-MNIST dataset, proving that MLPEPS has better performance than existing tensor networks; moreover, the accuracy of MLPEPS on the test set exceeds that of several existing neural networks. In addition, we verify the generalization ability of MLPEPS on the more complex COVID-19 Radiography dataset. The results prove that MLPEPS has better performance on real datasets than that of single-layer PEPS. However, its generalization ability is not as good as that of mature neural networks. MLPEPS does achieve accuracy close to that of the neural networks with a far smaller number of parameters. Thus, as a new machine learning method, MLPEPS shows great potential for image classification tasks.

The biggest advantage of MLPEPS is that it can be composed of different structures by modifying the size of the PEPS blocks; in future research, we intend to further explore the application of MLPEPS with different structures. Our experiments suggest that the use of different structures may have a very large impact on MLPEPS results. Therefore, the best MLPEPS structures for different tasks is a topic that can be further explored. In addition, deeper PEPS structures currently have relatively poor generalization ability. Therefore, there is a need to study ways of avoiding overfitting in order to improve generalization ability with more PEPS layers. This could be achieved by using the regularization mechanism from deep learning. Ideas from well-known deep neural networks can be applied to MLPEPS, which could constitute residual MLPEPS, dense MLPEPS, etc. Recently, the development of portable devices has made lightweight networks [43] a very important research direction. In this domain, MLPEPS can be used to compress the parameters of lightweight networks by taking advantage of its very low parameter requirements.

**Author Contributions:** Conceptualization, L.L. and H.L.; methodology, L.L. and H.L.; software, L.L.; validation, L.L. and H.L.; formal analysis, L.L. and H.L.; investigation, L.L. and H.L.; resources, L.L. and H.L.; data curation, L.L. and H.L.; writing—original draft preparation, L.L.; writing—review and editing, L.L. and H.L.; visualization, L.L.; supervision, L.L.; project administration, L.L. and H.L.; funding acquisition, H.L. All authors have read and agreed to the published version of the manuscript.

**Funding:** L.L. and H.L. have been supported by the National Natural Science Foundation of China (No.61702427) and the General Program of Chongqing Natural Science Foundation (No.CSTB2022NSCQ-MSX0749) the Venture & Innovation Support Program for Chongqing Overseas Returnees (No.cx2018076).

**Institutional Review Board Statement:** Not applicable.

**Informed Consent Statement:** Not applicable.

**Data Availability Statement:** Not applicable.

**Conflicts of Interest:** The authors declare no conflict of interest.

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
