# Peer review of "Multi-Layered Projected Entangled Pair States for Image Classification"

_sustainability, doi:10.3390/su15065120_

Round 1
Reviewer 1 Report (Previous Reviewer 1)
Title: Multi-Layered Projected Entangled Pair States for Image Classification
This manuscript is still lagging with sufficient content and presentation. Authors may publish this manuscript in conference proceedings not in a reputed journal like sustainability.
This manuscript has not presented proper literature review to claim multi-layered projected entangled based image classification.
Poor presentation of implementation details and results of observation.
Discussion for similar and opposite findings are lagging.
The performance measures are not sufficient if you use only accuracy.
Novelty of your work is missing. Where is the novelty of your work discussed?
Author Response
Please see the attachment.

Reviewer 2 Report (Previous Reviewer 3)
The authors have applied the corrections on article that it can be accepted for publication on JOURNAL
Author Response
Thank you very much for your comments.
Reviewer 3 Report (Previous Reviewer 2)
Modification completed, acceptable
Author Response
Thank you very much for your comments.
Reviewer 4 Report (New Reviewer)
This article proposes a hierarchical PEPS tensor network MLPEPS (Multilayer Projected Entangled Pair States) tensor network for image classification tasks. For the proof of concept, they proposed a mathematical model and experimented by performing classification tasks on the Fashion-MNIST dataset leading to acceptable results.
The authors should add a section in their introduction about their proposed method’s contributions, describe their experimental environment, and mention their plans for future work at the end of their conclusion. (Please refer to my comments on the document.)

Author Response
Please see the attachment. Thank you very much for your comments.

This manuscript is a resubmission of an earlier submission. The following is a list of the peer review reports and author responses from that submission.
Round 1
Reviewer 1 Report
The idea of this manuscript is good. However, I have some queries on your manuscript.
1. Abstract can be rewritten by introducing your proposed work’s performance measures.
2. construct block diagram / architecture diagram to represent the proposed MLPEPS.
3. Overall, your work is lagging with material and methods and detailed results.
4. Discussion about your result’s similar finding and dissimilar finding are missing.
5. You can study and write detailed literature reviews for PEPS.
6. Compare and contrast MLPEPS and PEPS performances.
7. Represent the hyper parameters introduced for your work towards capability of your work.
Author Response
- I've revised the Abstract section to focus on performance comparisons of experimental results.
- In Figure3 I have added diagram of PEPS block.
- I optimized in the methods and experimental results section, and really hope to get more specific Suggestions in the materials, methods and detailed results.
- In the Introduction section I include an introduction and discussion of the related methods.
- In the third section, there are short reviews of the PEPS classifier.
- In both experiments, I have added the performances comparison of PEPS and MLPEPS.
- In each experiment I added an introduction to hyperparameters like learning rate.
All revisions can be viewed in the attachment. Thanks a lot for your comments.

Reviewer 2 Report
This paper studies Projected Entangled Pair States (PEPS), and uses the idea of neural network to extend single-layer PEPS to multi-layer and apply it to image classification. The model in this study is tested on the Fashion MNIST dataset, which proves that MLPEPS has better performance than existing tensor networks, and the precision of the test set exceeds some neural networks. Later, it was tested on the COVID-19 Radiography dataset. Compared with some famous neural networks, there are still some gaps, but tensor networks have greater potential as a new machine learning method.
The article is logical, but there are still some format problems.
In Section 2 and 3, after the "defined as" in the formula definition, some have added colons and some have not, which is inconsistent. The author still needs to carefully check.
In Section 3.1, the reference after the formula definition should not need to be added, such as reference 8, otherwise it will be redundant.
The font size of the text in Figure 5 is too large. It should be adjusted to the same size as the text. The same problem exists in Figure 6.
In addition, it would be better if the reference of the picture in the text was closer to the picture, which is more convenient to read, such as Figure 2
For the tables in Chapter 4, the results with better model effects can be bolded, not limited to the best results in the proposed model.
To sum up, this study shows great potential in image classification as tensor networks, but there are still many details of the article that need to be revised. I hope the author will consider publishing after detailed revision.
Author Response
- The reference and colons after the formula have all been modified to the correct format.
- The size of the text in all figures has been modified to the text size.
- I've repositioned all figures and tables so they are closer to where they are cited for easier reading.
- For the tables in Chapter 4, the model with the best result has also been bolded.
All revisions can be viewed in the attachment. Thanks a lot for your comments.

Reviewer 3 Report
The authors have investigated a tool of Multi-layered PEPS (MLPEPS) tensor network model for image classification applied to COVID-19 Radiography datasets and clothes.
The manuscript appears well- structured and clear . The english needs to be improved in order to easy interpret and understand the article ; for example in
In order to better obtain the global information of the picture, the Projected Entangled Pair States (PEPS) tensor network [8] with the same geometric structure as the natural image is introduced into supervised learning,…….
In order to obtain better….the projected ..is introduced with the same geometric…
I suggest to extend the introduction considering to add/cite the methodology of classification well-studied in :
Small lung nodules detection based on fuzzy-logic and probabilistic neural network with bioinspired reinforcement learning
G Capizzi, GL Sciuto, C Napoli, D Połap, M Woźniak
IEEE Transactions on Fuzzy Systems 28 (6), 1178-1189
Organic solar cells defects detection by means of an elliptical basis neural network and a new feature extraction technique
GL Sciuto, C Napoli, G Capizzi, R Shikler
Optik 194, 163038
Author Response
I've added a reference to the suggested method in the second paragraph of the Introduction, see references 12 and 13. Please see the attachment.
Thanks a lot for your comments!

Round 2
Reviewer 1 Report
Title: Multi-Layered Projected Entangled Pair States for Image Classification
-
This article has serious article writing issues. For example I can see “Matrix product states (MPS)” both abstract as well as in INtroduction. Then no meaning for abbreviation.
-
Most of the comments are not incorporated in the old article. Better revise this manuscript and send it back.
-
Author response file for previous review (round 1) is not found as attachment, instead they have uploaded an old article.
Round 3
Reviewer 1 Report
Title: Multi-Layered Projected Entangled Pair States for Image Classification
-
This article once again has serious issues. For example, Tree tensor networks (TTN) abbreviated and once again used Tree tensor networks again. Then what is the use of abbreviation? Authors are asked to review this issue seriously.
-
The authors did not conduct any proper related work before considering your idea for image classification. High standard technical papers should possess related works.
-
Detailed experimental works must be written including block diagrams of flow of work.
-
Comparative results and discussion are missing in the results part. Better keep a graphical comparative graph with existing works’ results with your proposed work results.
Round 4
Reviewer 1 Report
Title: Multi-Layered Projected Entangled Pair States for Image Classification
1. I have read the short passage from your article. Can you classify whether the sec. Or simply sec.
The Sec.2 presents methods for performing supervised learning using tensor networks. Sec.3 details the method for constructing MLPEPS. The Sec.4 introduces the results of MLPEPS on image classification and compares it with other methods. Sec.5 concludes the paper and discusses more ways to use MLPEPS.
2. Once again, the content required for introduction and related works are not sufficient at al. you can go through more related references and cite them. If possible present a comparative table alongside their methods, limitation, advantage etc.
